# Evaluation of plasma viral-load monitoring and the prevention of mother-to-child transmission of HIV-1 in three health facilities of the Littoral region of Cameroon

Etienne Verlain Fouedjio Kafack[1], Joseph Fokam[1,2]*, Theophile Njamen Nana[1], Arthur Saniotis[3,4‡], Gregory Edie Halle-Ekane[1,5‡]*

1 Faculty of Health Science, University of Buea, Buea, Cameroon, 2 Virology Laboratory, Chantal BIYA International Reference Centre for Research on HIV/AIDS Prevention and Management, Yaounde, Cameroon, 3 Bachelor of Doctor Assistance Department, DDT College of Medicine, Gabarone, Botswana, 4 Biological Anthropology and Comparative Anatomy Research Unit, School of Biomedicine, University of Adelaide, Adelaide, Australia, 5 Douala General Hospital, Douala, Cameroon

☯ These authors contributed equally to this work.
‡ AS and GEHE also contributed equally to this work as senior authors.
* halle-ekane.edie@ubuea.cm (GEHE); josephfokam@gmail.com (JF)

**Data Availability Statement:** Data are all contained within the paper.

## Abstract

### Background

Prevention of mother-to-child transmission (PMTCT) has reduced HIV incidence among new-borns. However, PMTCT remains concerning in sub-Saharan Africa due to bottlenecks including viral load (VL) monitoring during pregnancy. We assessed VL coverage and materno-foetal outcomes of pregnancy among HIV-infected women within the Cameroonian context.

### Methods

A hospital-based study was conducted among HIV-infected mothers and their babies in three facilities of the Littoral region of Cameroon from January 2019 to May 2021. Maternal VL-coverage was monitored during pregnancy (VL>1000 copies/ml or unknown were classified as MTCT high-risk group); HIV early infant diagnosis (EID) was evaluated by PCR at six-weeks after birth, and EID results were analysed according to maternal VL; $p<0.05$ was considered statistically significant.

### Results

Of 135 HIV-infected pregnant women enrolled (median [IQR] age 39 [27–37] years), VL-coverage during antenatal care (ANC) was 50.4% (68/135), with a lower VL-coverage in 2019 (37.5% vs. 61.9%, p = 0.0069). Married women vs. single (61.8% vs. 42.5%, p = 0.0275) and those on treatment before vs. during pregnancy (56.7% vs. 5.8%, p = 0.0043) had a higher VL-coverage, respectively. Among those with known VL, 10.3% (7/68) had high (VL>1000 copies/mL), 22.1% (15/68) had low (50–1000 copies/mL), and 67.6% (46/68) had undetectable (<50 copies/mL) VL, suggesting an overall viral suppression

**Funding:** The study was supported by the government of Cameroon through provision of viral load testing kit and antiretroviral drugs to all the study participants. "The funders had no role in study design, data collection and analysis, decision to publish, or preparation of the manuscript.

**Competing interests:** The authors have declared that no competing interests exist.

(<1000copies/mL) of 89.7% (61/68). Vaginal delivery was 80.75% (109/135) regardless of VL, including 81.1% (59/74) women in the high-risk group. EID coverage was 88.1% (119/135) and the rate of HIV-1 MTCT was 1.68% (2/119). Both HIV-positive infants were from the high-risk group, had prolonged labour, had vaginal delivery and were breastfed.

## Conclusion

In these Cameroonian settings, VL-coverage remains suboptimal (below 90%) among ANC attendees, and women at high-risk of MTCT mainly have vaginal delivery. Viral suppression rate remains below the target (below 90%) for accelerating the elimination of MTCT. HIV-MTCT persists, and might be driven essentially by poor VL monitoring. Thus, achieving an optimal PMTCT performance requires a thorough compliance to virologic assessment during ANC.

## Introduction

In spite of the global progress in maternal and child health, HIV/AIDS remains a threat for pregnant women with higher prevalence reported among target populations living in sub-Saharan Africa (SSA) [1, 2]. This disproportionate burden placed on women indicates a substantial number of HIV-exposed children in SSA, which result in high rates of new HIV infections and HIV-related illness and death [3]. Similarly, pregnant women in Cameroon are also disproportionately infected with HIV (national prevalence of 4.26% [IC95%: 3,79–4,79]) compared with the general population (national prevalence of 2.7%). Of note, in the Littoral region of Cameroon, the proportion of HIV-infected women of reproductive age is 3.5 times more than that of men (due to their known vulnerability to infection), suggesting a substantial risk of HIV transmission to new-borns in the absence of specific maternal and child healthcare interventions within this region [4].

Prevention of Mother-To-Child Transmission (PMTCT) program remains a priority for HIV-infected pregnant women and their babies in ensuring a generation of HIV-free infants while keeping the mothers alive. In this frame, strategies have been developed to reduce mother-to-child transmission (MTCT) which include the use of antiretroviral drugs, the mode of delivery related to viral load (VL) in the antepartum period, and breast milk substitutes as measures to promote PMTCT [5]. These strategic interventions are of paramount importance in curving down the current high burden of HIV vertical transmission in SSA, reported with a cumulative *in utero*, intrapartum, and postpartum MTCT of 25–35% [3]. Of relevance, maternal VL is by far the most predictive factor for perinatal HIV transmission, as higher HIV VLs correlate with a greater risk of perinatal transmission [6]. Within the frame of PMTCT, the goal of antiretroviral therapy (ART) is to suppress the maternal VL in order to lower the risk of transmitting the virus to their infant during pregnancy, delivery and breastfeeding [7]. Since plasma VL level is classified as the main driver of HIV-1 MTCT, the World Health Organization (WHO) has recommended routine VL testing since 2016 for pregnant women living with HIV [8], with the ultimate objective of achieving maternal viral suppression before delivery, and sustaining the viral control through breastfeeding and beyond [9]. In Cameroon, routine VL has been endorsed as the reference marker for treatment monitoring and for eliminating MTCT following the WHO recommendations. Considering the financial, logistics

and operational challenges in expanding VL testing for HIV-positive populations in SSA countries [9–12], scale-up of VL needs to be closely monitored within the frame of the entire PMTCT cascade care in Cameroon. This is important in order to identify bottlenecks during implementation and propose evidence-based measures for an optimal PMTCT performance. So far, little is known about the effective implementation of VL monitoring as per the WHO-recommended PMTCT guidelines and the potential effect of VL monitoring on materno-foetal outcomes of pregnancy among women living with HIV in the Cameroonian context.

We, therefore, attempted to ascertain the performance in plasma VL coverage during ANC, to describe the mode of delivery of HIV infected mothers, and to estimate the rate of HIV-1 vertical transmission according to maternal VL.

## Methods

### Study design

A hospital-based study was conducted among HIV-infected pregnant women and their newborns at three facilities (*Douala General Hospital*, *Laquintinie Hospital*, *and Nkongsamba Regional Hospital*) in the Littoral region of Cameroon from January 2019 to May 2021. As per the routine national guidelines for PMTCT in Cameroon, pregnant women in each study site were monitored from their first antenatal care (ANC) attendance until delivery. Also, infants were followed-up to six weeks postnatal for HIV early infant diagnosis (EID).

### Description of the study setting

In Cameroon, the health care landscape is structured according to three levels (central, intermediate and peripheral) and three sub-levels: public sub-sector; private sub-sector (non-profit making and for-profit) and traditional sub-sector. Each level of the pyramid consisted of administrative, health and dialogue levels. Peripheral level facilities (district hospitals, clinic, health centre) focus on primary care and community health. Intermediate level facilities (3rd category hospitals: regional hospitals and equivalent) provide more specialized services, including HIV care and maternity services, while central level facilities (1st and 2nd category hospitals: General hospital, university teaching hospital and equivalent) provide highly specialized services, in addition to HIV care and maternity services.

PMTCT guidelines in Cameroon, used in the study sites, recommend early ANC attendance (eight visits recommended during pregnancy), antiretroviral therapy (ART) initiation as early as possible, and routine VL monitoring of pregnant women [13]. According to the national guidelines, schedule for VL monitoring differs according to the moment of HIV diagnosis during pregnancy. Briefly: (a) for pregnant women previously known to be HIV-positive and already on ART, baseline VL measurement is recommended at the first ANC attendance (i.e. if previous VL was performed more than one month ago) then quarterly VL measurements; (b) for pregnant women newly diagnosed with HIV and who have commenced ART, baseline VL measurement is recommended at 3 months after ART initiation, with quarterly VL measurements (last VL at week 32–36 of amenorrhea).

### Study facilities

The three sites selected for the study were all in the Littoral region of Cameroon, the sites ranged from the 1st category to 3rd category hospitals; all the facilities were government hospitals; two facilities (*Douala General Hospital and Laquintine Hospital*) were in a urban setting while the other (*Nkongsamba Regional Hospital*) was in semi-urban setting (Table 1).

**Table 1. Characteristics of the selected health facilities.**

| Name of the health facility | Health care level | Category Hospital | Location | Laboratory in charge of VL test |
|---|---|---|---|---|
| Douala General Hospital | Central | First | Douala | External to hospital |
| Laquintinie Douala Hospital | Central | Second | Douala | Inside the hospital |
| Nkongsamba Regional Hospital | Intermediate | Third | Nkongsamba | External to hospital |

In these facilities, VL measurement is recommended at the first ANC attendance (i.e. if previous VL was performed more than three months ago) or three months after initiation for newly diagnosed HIV infected pregnant women, then quarterly VL measurements (last VL at week 32–36 of amenorrhea).

## Sampling technique and sample size calculation

A consecutive and non-probabilistic sampling method was used to enrol study participants who meet the inclusion criteria (pregnant women who is HIV-seropositive, receiving antiretroviral therapy as per our national guidelines, registered for ANC in one of the study sites, and has provided a written informed consent). Pregnant women newly diagnose HIV-infected during labour and those eligible but with incomplete file were excluded.

The minimum sample size was calculated using the Cochran's Formula as follows:

$$n = \frac{(z)^2 p(1-p)}{d^2}$$

With n being the minimum sample size; p being the proportion of viral load coverage among HIV-infected pregnant women in Cameroon during the year 2021 (90.7%, i.e. 0.907) [13]; with z being the standard normal random variable at 95% confidence interval (1.96); and d being the error margin (5%, i.e. 0.05). With n = 129.6, the minimum sample size for the study was 130 pregnant women to be enrolled.

## Enrolment of study participants

At the study sites, a written informed consent was first provided by every eligible woman (pregnant and living with HIV), followed by an interview using a structured questionnaire. All eligible women were scheduled for two assessments (the first assessment was after delivery and the second was at six weeks postnatal for the infant's HIV status). Medical records of these study participants were thoroughly reviewed for the collection of study variables.

## Data collection

Following ethical clearance and administrative authorizations, pregnant women were approached and received the study information sheet, after an interview for clarity on the study goals, these women were enrolled if eligible and scheduled for a first assessment after delivery and a second six weeks postnatal. Additionally, medical records of eligible pregnant women who were in labour and delivered in our study site during the study period were reviewed. Data were double-checked to assess completeness, and missing data were completed wherever necessary.

Data were collected after delivery and six weeks after follow-up using the data collection sheet. The data collection sheet included the following sections: VL measurements, mode of delivery according to VL and early infant diagnosis result at six weeks (primary outcomes). Socio-demographic and clinical data, feeding option, birth weight, and nevirapine prophylaxis

were considered as secondary outcomes. Routine drug compliance was assessed based on patient reports and physician's reports in the hospital book. Patients with high (VL>1000 copies/mL) or unknown VL were classified as high-risk group.

## Data analysis

All VL results during pregnancy were included in analysis. We assessed the proportion receiving guidelines-adherent VL measurement and factors associated with VL measurement. Frequencies and percentages were computed for categorical variables. Mean and standard deviation for continuous variables were tested. Materno-foetal outcomes were primarily referring to the VL (for the mother) and to the HIV EID status (for the infant).

Chi's square or Fisher's exact test where applicable were used to compare categorical variables. A bivariate analysis was carried out to determine associations with all predictor variables with a p-value ≤ of 0.05. Logistic regression models were fitted to control for confounders. Factors associated with VL coverage and HIV transmission (EID result) were analysed according to the mode of delivery, the level of health facility, socio-demographic, clinical and laboratory parameter (maternal VL); with p-value <0.05 considered as statistically significant.

## Ethical considerations

Prior to study commencement, ethical clearance was obtained from the Institutional Review Board of the Faculty of Health Sciences, University of Buea (Ref. No. *2021/1307-02/ UB/SG/ IRB/FHS)*. Administrative approval was obtained from the Managing Director of the Nkong-samba Regional Hospital (Ref. No. *058/AR/MINSANTE/DRSPL/HRN/21*), Laquintinie Douala Hospital (Ref. No. *0796/AR/MINSANTE/DHL/CM*) and Douala General Hospital (Ref. No. *69AR/MINSANTE/HGD/DM/02/21*).

For each study participant, written informed consent was provided; confidentiality in handling the participant information was ensured by the use of specific identifiers; the storage of paper sheets in a looked office, and data entry in an encrypted database on a password-protected computer, and clinical advice or medical care was provided individually to each participant for their personal health benefits.

## Results

### Socio-demographic characteristics of the study population

A total of 135 HIV-infected pregnant women were enrolled as participants; the median [IQR] age was 39 [27–37] years. The majority of participants (61.5%) were in the age group 31–40 years and 65.9% (89/135) of them had a secondary-level of education. Regarding marital status, more than half (58.5%) of these pregnant women were single, and the majority (95.6%) were Christians.

### Clinical information of the study participants

All the participants were ANC attendees and 69.6% (94/135) were multiparous. The average ANC visit was 2.76±1.33SD and most of the participants 106 (78.5%) started antenatal visits during the second trimester. Furthermore, among pregnant women living with HIV, 125 (92.6%) mostly consulted obstetricians for their ANC visits (Table 2).

The majority, 87.4% (118/135) of these ANC attendees knew their HIV status before pregnancy had commenced ART before the current pregnancy. Among them, 129 (95.6%) were still on the first-line ART regimen, and 94.8% (128) were classified as WHO clinical stage I.

Table 2. Obstetrics information of study population (N = 135).

| Variables | DGH[a] (N$_1$ = 27) n (%) | LDH[b] (N2 = 63) n (%) | NRH[c] (N3 = 45) n (%) | Total (N = N$_1$+N$_2$+N$_3$ = 135) n (%) |
|---|---|---|---|---|
| **Parity** | | | | |
| Primiparous | 2 (7.4) | 6 (9.5) | 7 (15.6) | 15 (11.1) |
| Multiparous | **18 (66.7)** | **47 (74.6)** | **29 (64.4)** | **94 (69.6)** |
| Grand multiparous (parity≥ 5) | 7 (25.9) | 10 (15.9) | 9 (20.0) | 26 (19.3) |
| **Numbers of ANC done** | | M±SD (2.76±1.33) | | |
| < 4 | **15 (55.5)** | **58 (92.0)** | **42 (93.4)** | **115 (85.2)** |
| 4–8 | 8 (29.7) | 3 (4.8) | 2 (4.4) | 13 (9.6) |
| > 8 | 4 (14.8) | 2 (3.2) | 1 (2.2) | 7 (5.2) |
| **Initiation of ANC** | | | | |
| 1st Trimester | 10 (37.0) | 4 (6.3) | 9 (20.0) | 23 (17.0) |
| 2nd trimester | **16 (59.3)** | **55 (87.3)** | **35 (77.8)** | **106 (78.5)** |
| 3rd trimester | 1 (3.7) | 4 (6.3) | 1 (2.2) | 6 (4.4) |
| **Obstetric care giver** | | | | |
| Obstetrician | **27 (100.0)** | **53 (84.1)** | **45 (100.0)** | **125 (92.6)** |
| General practitioner | 0 (0.0) | 5 (7.9) | 0 (0.0) | 5 (3.7) |
| Midwife/Nurse | 0 (0.0%) | 5 (7.9%) | 0 (0.0) | 5 (3.7) |

[a]Douala General Hospital;

[b]Laquintinie Hospital Douala;

[c]Nkongsamba Regional Hospital; Bold number represent the highest proportion.

Looking at the obstetrics information of HIV-infected pregnant women, 92.6% (125/135) had spontaneous labour and the mean gestational age at labour was 38,55±1.74SD; duration for rupture of membranes (ROM) was normal for 91.1%; and the population of HIV-exposed children were composed of more male than female (55.6% vs 44.4% respectively) and 89.6% had a normal weight (Table 3).

## Factors associated with viral load coverage and HIV vertical transmission

Of the 135 enrolled HIV-infected pregnant women, only 68 had at least one plasma viral load assessed during pregnancy, which gave a VL-coverage during antenatal care of 50.4% (68/135). Among those with known VL, 10.3% (7/68) had high (VL>1000 copies/mL), 22.1% (15/68) had low (50–1000 copies/mL), and 67.6% (46/68) had undetectable (<50 copies/mL) VL, which resulted in an overall viral suppression rate of 89.7% (61/68) (Table 4).

Among those for whom a PVL was successfully performed, the mean number of PVL measurement done during pregnancy was 1.03±0.171 SD; only 2/68 (3.0%) had a repeated VL after the baseline VL during the ANC period; and most VL tests (41.2%) were done during the 3rd trimester of pregnancy. Those on ART before pregnancy had a significantly higher proportion of VL coverage during ANC as compared to those diagnosed HIV-positive during the present ANC (56.7% versus 5.8% respectively, AOR = 21.270, p = 0.0043). Married women had a significantly higher proportion of VL-coverage as compared to single (61.8% versus 42.5% respectively, AOR = 2.385; p = 0.0275). According to the year of enrolment, participants enrolled before the year-2020 (year-2019) had a significantly lower VL coverage as compared to those enrolled thereafter (37.5% v. 61.9%, AOR = 0.351; p = 0.0069), suggesting a scale-up in VL measurement overtime (Table 5).

Approximately 81% (109/135) of the participating pregnant women living with HIV delivered vaginally, including 81.1% (59/74) women in the high-risk group. During the study

**Table 3. Characteristics of labour, delivery and new-borns.**

| Variables | DGH[a] (N₁ = 27) n (%) | LDH[b] (N2 = 63) n (%) | NRH[c] (N3 = 45) n (%) | Total (N = N₁+N₂+N₃ = 135) n (%) |
|---|---|---|---|---|
| **Labour** | | | | |
| Spontaneous | **26 (96.3)** | **59 (93.7)** | **40 (88.9)** | **125 (92.6)** |
| Induction | 1 (3.7) | 4 (6.3) | 5 (11.1) | 10 (7.4) |
| **Gestational age at labour** | | | M±SD (38.55±1.74) | |
| < 37 weeks | 3 (11.1) | 3 (4.8) | 5 (11.1) | 11 (8.2) |
| ≥ 37 weeks | **24 (88.9)** | **60 (95.2)** | **40 (88.9)** | **124 (91.8)** |
| **Duration Labour** | | | | |
| ≤20hrs | **27 (100.0)** | **58 (92.1)** | **38 (84.4)** | **123 (91.1)** |
| >20hrs | 0 (0.0) | 5 (7.9) | 7 (15.6) | 12 (8.9) |
| **Duration ROM** | | | | |
| Normal (≤12hrs) | **22 (81.5)** | **59 (93.7)** | **42 (93.3)** | **123 (91.1)** |
| Prolonged (>12hrs) | 5 (18.5) | 4 (6.3) | 3 (6.7) | 12 (8.9) |
| **Mode of delivery** | | | | |
| Vaginal birth | 21 (77.8) | 50 (79.4) | 38 (84.4) | **109 (80.7)** |
| Caesarean Section | 6 (22.2) | 13 (20.6) | 7 (15.6) | 26 (19.3) |
| **Characteristics of new-borns** | | | | |
| **Sex of Exposed child** | | | | |
| Male | **15 (55.6)** | **37 (58.7)** | **23 (51.1)** | **75 (55.6)** |
| Female | 12 (44.4) | 26 (41.3) | 22 (48.9) | 60 (44.4) |
| **Weight of exposed child (g)** | | | M±SD (3112.5±480.19) | |
| <2500 | 3 (11.1) | 7 (11.1) | 2 (4.4) | 12 (8.9) |
| 2500–4000 | **22 (81.5)** | **56 (88.9)** | **43 (95.6)** | **121 (89.6)** |
| >4000 | 2 (7.4) | 0 (0.0) | 0 (0.0) | 2 (1.5) |
| **Infant nutrition** | | | | |
| Breastfeeding | 11 (40.7) | **61 (96.8)** | **44 (97.8)** | **116 (85.9)** |
| Formula feeding | **14 (51.9)** | 2 (3.2) | 1 (2.2) | 17 (12.6) |
| Both | 2 (7.4) | 0 (0.0) | 0 (0.0) | 2 (1.5) |

[a]Douala General Hospital;

[b]Laquintinie Hospital Douala;

[c]Nkongsamba Regional Hospital; ROM: rupture of the membrane; in bold are high proportions and numbers.

period, HIV-1 EID coverage was 88.1% (119/135); and the rate of HIV-1 MTCT was 1.68% (2/119). There was no statistically significant difference between the rate of MTCT and the mode of delivery, plasma VL and the category of hospitals.

Bivariate analysis of determinants of MTCT revealed that HIV-exposed infants without nevirapine prophylaxis (p = 0.0028), prolonged labour of ≥ 12hours (p = 0.0129) and poor maternal adherence to cART (p = 0.0019) appeared to be associated with a positive EID result. Inversely, after multivariable analysis adjusting for possible confounders, none of the study parameters was found to be associated with MTCT of HIV (see Table 6).

## Discussion

Due to persisting HIV vertical transmission SSA, several countries in this region remain on the priority list for PMTCT interventions, including Cameroon. Thus, identifying barriers for achieving effective elimination of HIV MTCT remains a public health priority. This is particularly true in SSA where only 35–50% of pregnant women on lifelong ART regimens have been

**Table 4. Evaluation of plasma viral load coverage among HIV-infected pregnant women.**

| Variables | | NRH | LDH | DGH | Total |
|---|---|---|---|---|---|
| | | n (%) | n (%) | n (%) | n (%) |
| **Plasma viral load testing during ANC** | | | | | |
| No | | **25** (55.6) | 28 (44.4) | **14** (51.9) | 67 (49.6) |
| Yes | | 20 (44.4) | **35** (55.6) | 13 (48.1) | **68** (50.4) |
| Total | | 45 (100) | 63 (100) | 27 (100) | 135 (100) |
| **Plasma viral load testing during ANC according to year** | | | | | |
| <2020 | No | **12** (50) | **19** (70) | **9** (69) | 40 (62.5) |
| | Yes | **12** (50) | 8 (30) | 4 (31) | **24** (37.5) |
| ≥2020 | No | **13** (61.9) | 9 (25) | 5 (35.7) | 27 (38) |
| | Yes | 8 (38.1) | **27** (75) | **9** (64.3) | **44** (62) |
| **Numbers of plasma viral load testing done during ANC** | | | | | |
| | | M±SD (1.03±0.171) | | | |
| 01 | | **19** (95.0) | **34** (97.1) | **13** (100.0) | **66 (97.0)** |
| 02 | | 1 (5.0) | 1 (2.9) | 0 (0.0) | 2 (3.0) |
| 03 | | 0 (0.0) | 0 (0.0) | 0 (0.0) | 0 (0.0) |
| Total | | 20 (100) | 35 (100) | 13 (100) | 68 (100.0) |
| **Numbers of Plasma viral load testing in the 3rd Trimester** | | | | | |
| 3rd Trimester | | **7** (35) | **15** (42.85) | **6** (46.15) | **28 (41.2)** |
| Others | | 13 (65) | 20 (57.15) | 7 (53.85) | 40 (58.8) |
| **Plasma viral load level in the 3rd Trimester** | | | | | |
| < 1000copies/mL | | 6 (85.7) | 15 (100.0) | 5 (81.33) | **26 (92.87)** |
| ≥ 1000copies/mL | | 1 (14.3) | 0 (0.0) | 1 (16.67) | 2 (7.13) |
| **Plasma viral load level general population** | | | | | |
| > 1000copies/mL | | 3 (6.7) | 2 (3.2) | 2 (7.4) | 7 (5.18) |
| 50–1000 copies/mL | | 4 (8.9) | 9 (14.3) | 2 (7.4) | 15 (11.11) |
| < 50 copies/mL | | 13 (28.9) | 24 (38.1) | 9 (33.3) | 46 (34.07) |
| Unknown value | | **25** (55.5) | **28** (44.4) | **14** (51.9) | 67 (49.6) |

tested for VL [9, 14–16]. In Cameroon where HIV in pregnancy remains substantially high (about 4%) [3, 4], there is need of evidence on VL coverage and its possible effect on HIV MTCT, for timely measures toward optimal elimination of MTCT at country-level.

Our data show that approximately 50% of the women in these three Cameroon facilities received a VL test during PMTCT, with only 3% having a repeated VL before delivery. As this finding is similar to the uptake levels reported previously, the current situation underscores the fact that only one in every two pregnant women have access to VL measurement in the course of pregnancy in SSA countries [9, 14–16]. Unlike recommendations from our national

**Table 5. Factors associated with plasma viral load testing.**

| | | 95% C.I. | | |
|---|---|---|---|---|
| Term | AOR[a] | MIN | MAX | P-Value |
| HIV Diagnosis or ART initiation (Before Pregnancy/During Pregnancy) | 21,270 | 2,613 | 173,139 | **0,0043***  |
| Married status (married/single) | 2,385 | 1,101 | 5,164 | **0,0275*** |
| Period year (Before 2020/From 2020) | 0,351 | 0,164 | 0,750 | **0,0069*** |

[a]Adjusted Odd ratio.

*Statistically significant value.

**Table 6. Bivariate and multivariable analysis of determinants of MTCT[a].**

|  | OR[b] | CI[c] | p-Value | AOR[d] | CI | p-value |
|---|---|---|---|---|---|---|
| **Infant prophylaxis** |  |  |  |  |  |  |
| No | 1 | 6.53–8455.77 | **0.0028**[*] | $1.615^{E}9$ | 0.0- $>1^{E}12$ | 1.000 |
| Yes | 235.0 |  |  |  |  |  |
| **Duration of Labour (hours)** |  |  |  |  |  |  |
| $\geq 12$ | 1 | 2.30–1138.42 | **0.0129**[*] | 0.000 | 0.0 - $>1^{E}12$ | 0.996 |
| $< 12$ | 51.19 |  |  |  |  |  |
| **Maternal adherence to cART** |  |  |  |  |  |  |
| Poor | 1 |  |  |  |  |  |
| Good | 163.57 | 6.55–4082.82 | **0.0019**[*] | 0.000 | 0.0 - $>1^{E}12$ | 0.995 |

[a]Mother-to-child transmission.

[b]Odd ratio.

[c]Confidence Interval.

[d]Adjusted odd ratio.

[*]Statistically significant value.

guidelines on PMTCT which recommend three plasma viral load testing during pregnancy, poor VL monitoring in these settings emphasises the need for greater awareness among health practitioners regarding correct PMTCT practices and a call for programmers to ensure a wider placement of VL devices at PMTCT sites in the country. Our findings are different from a study conducted in Kenya where a higher rate was reported, though still below standards (70% of HIV pregnant women did VL test and 33% of repeat VL test during ANC) [9].

In the current era of PMTCT option B+ in Cameroon, few pregnant women (less than 10% in the case of our study) are newly diagnosed, indicating that hospital-based PMTCT program is highly dominated by experienced clients, thus calling for strategies to track missing HIV-infected mothers at community-level. On the other hand, even though the newly diagnosed mothers are enrolled on ART, very few received a PVL test during pregnancy, thus considering such clients as at high risk of HIV vertical transmission [13]. Additionally, there was no participant in this category with a repeated test, while only 2% of women with prior knowledge of their HIV status had a repeat VL test during pregnancy. Thus, while reinforcing PMTCT monitoring during ANC, a special emphasis should be given to women who have been recently diagnosed with HIV infection, in order to limit the current poor virologic monitoring, thereby, mitigating the risk of MTCT in this high-risk population. In fact, those who were previously provided treatment were more likely than those newly diagnosed to receive a PVL test. This finding is also justified by the fact that most health care providers were still applying the guideline of managing HIV-infection in the general population which stipulated that newly diagnosed HIV-infected patients should receive a VL test six months after initiation of therapy. There were also significant differences between the period before the year 2020 (<2020) and from the year 2020 ($\geq$ 2020), where married women were compared to single women. These findings are in line with the increasing scale-up in VL coverage overtime, which should be supplemented with continuous training and capacity-building of staff on PMTCT updates, as well as in developing specific counselling guidelines for ANC attendees on key periods of VL monitoring in PMTCT.

In our study settings, approximately 8 out of 10 pregnant women living with HIV had an unknown PVL or PVL> 1000copies/ml near delivery (3rd trimester). This finding is significant for drawing attention to virologic testing near delivery, as this parameter is crucial for

decision making on delivery mode. Of note, the ratio of vaginal delivery to caesarean section was 4 to 1, and differed from a study done in Philadelphia which found a ratio of vaginal delivery to c-section of 1 to 1 among HIV infected pregnant women [17]. In SSA settings where C-section is poorly perceived by the community, the lack of VL results have not yet been considered clinically as a potential high-risk of HIV vertical transmission. In contrast, in high-income countries, c-section is both promoted and accepted by ANC attendees as a preferred means to prevent perinatal transmission of HIV. This lack of knowledge and practice of c-section in Cameroon should incite strategies to reinforce PMTCT best practices in pregnant women living with HIV without a PVL test or a PVL≥1000copies/ml.

Furthermore, we observed that 8 in 10 MTCT high-risk mothers delivered through vaginal deliveries, which differed from Thompson *et al*. who reported slightly above 3 in 10 vaginal deliveries among the high-risk group [17]. Thus, while stressing in need to adherence to PMTCT interventions, VL should be considered at the earliest convenience to support interventions after delivery (i.e. infant prophylaxis duration, feeding option, etc) for the high-risk group.

Among participants who had a VL test at closer to delivery, vaginal delivery was the common mode of delivery for approximately 80% of participants. This finding is comparable to the mode of delivery of high-risk groups (80.75%) in other African settings, regardless of plasma viral load or category of the health facility hospital, that falls in line with current recommendations. However, Egbe *et al*., as well as some obstetricians proposed surgical techniques to reduce MTCT of HIV by using c-section [18], or by considering the VL near delivery time as an indicator of the appropriate mode of delivery (as currently applied in high incomes countries) [17].

Regarding EID coverage, 2 out of 10 early infant diagnosis (EID) results were unavailable within the study period, which indicates limitations in case management throughout the PMTCT cascade and barriers in achieving eMTCT. Among those with an available EID result, less than 2% of HIV-exposed infant were HIV-infected, indicating a good PMTCT performance. This low rate of MTCT of HIV at 6 weeks could be due to the effectiveness of option B + in our context, especially with the significant difference observed between the EID result and maternal adherence to ART (p = 0.001). There was also a significant difference between the EID result and the use of infant prophylaxis (p = 0.0168), supporting the relevance of these interventions in eliminating MTCT. Similar findings were reported by Thompson *et al*. in Philadelphia [17] and by the National AIDS control committee in Cameroon [4].

After statistical adjustments, there were no determinants of HIV MTCT in our study population, except an association found between HIV EID outcome and the use of nevirapine prophylaxis (p = 0.0168), poor maternal adherence to ART (p = 0.001) and duration of labour of >20 hours (p = 0.012). Thus, combining these clinical conditions to VL would contribute considerably in further reducing MTCT in Cameroon [1].

The current study had some potential limitations regarding the PMTCT program. For instance, given that it was a hospital-based study, it did not take into account those women who had home births. This would have had greater coverage by considering community-based vertical transmission of HIV, for better utility into PMTCT programs at a public-health level. Moreover, our focus was limited to MTCT at delivery, indicating possible new cases of MTCT thereafter (especially for breastfeeding infants). In spite of the high number of women of child bearing age and pregnant women in our setting, the sample size (n = 135 HIV-pregnant women enrolled) during the study period correlates to the reducing burden of HIV-infection in ANC [13]. Nonetheless, the multicentre design (03 levels of referrals hospital in Cameroon) gives room for evidence-based regimens that could be useful for the better implementation of PMTCT programs in both urban and rural settings of Cameroon.

## Conclusion

Within major PMTCT sites in the Littoral region of Cameroon, plasma VL coverage was poor, indicating a suboptimal VL monitoring among pregnant women living with HIV, which goes with a sustained risk of HIV vertical transmission. Nonetheless, there is an improvement in scaling-up VL overtime, underscoring the benefit of universal access to health care following the government guidelines for monitoring HIV treatment. The majority of deliveries for women with an unknown plasma VL or VL >1,000 copies/ml occurred via vaginal delivery, thus highlighting the persistent risk of HIV vertical transmission. Although the rate of early HIV MTCT appeared to be low, positivity among HIV-exposed infants seemingly occurs with poor maternal adherence to ART, lack of nevirapine prophylaxis, vaginal deliveries, high or unknown maternal VL, breastfeeding and prolonged labour.

## Acknowledgments

We thank the clinicians and nurses of the respective study sites for their contributions in the clinical monitoring during ANC, delivery and postnatal follow-up. We also thank the laboratory staffs for ensuring viral load measurement and HIV early infant diagnosis by PCR.

## Author Contributions

**Conceptualization:** Etienne Verlain Fouedjio Kafack, Joseph Fokam, Arthur Saniotis, Gregory Edie Halle-Ekane.

**Data curation:** Etienne Verlain Fouedjio Kafack, Joseph Fokam.

**Formal analysis:** Etienne Verlain Fouedjio Kafack.

**Investigation:** Etienne Verlain Fouedjio Kafack, Gregory Edie Halle-Ekane.

**Methodology:** Etienne Verlain Fouedjio Kafack, Joseph Fokam, Theophile Njamen Nana, Arthur Saniotis, Gregory Edie Halle-Ekane.

**Supervision:** Joseph Fokam, Theophile Njamen Nana, Gregory Edie Halle-Ekane.

**Validation:** Arthur Saniotis.

**Visualization:** Gregory Edie Halle-Ekane.

**Writing – original draft:** Etienne Verlain Fouedjio Kafack, Joseph Fokam.

**Writing – review & editing:** Joseph Fokam, Theophile Njamen Nana, Arthur Saniotis, Gregory Edie Halle-Ekane.

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
