## [Decision Letter · Decision Letter 0]

8 Jul 2022

PONE-D-22-15160Evaluation of Plasma Viral-Load Monitoring and the Prevention of Mother-To-Child Transmission of HIV-1 in the Littoral Region of CameroonPLOS ONE

Dear Dr. Fokam,

Thank you for submitting your manuscript to PLOS ONE. After careful consideration, we feel that it has merit but does not fully meet PLOS ONE’s publication criteria as it currently stands. Therefore, we invite you to submit a revised version of the manuscript that addresses the points raised during the review process.

We look forward to receiving your revised manuscript.

Kind regards,

Nafees Ahmad, Ph.D.

Academic Editor

PLOS ONE

Journal Requirements:

3. In your Methods section, please provide additional information on how the sample size was determined and why sample size calculation was not carried out before sample collection.

 NA. 

Additional Editor Comments:

There are some issues with the design of the study/cohort, as shown as a major concern by one of the reviewers, should be redesigned and addressed. The inclusion of the subjects should be in a systematic manner to provide meaningful conclusions.

Reviewers' comments:

Reviewer's Responses to Questions

**Comments to the Author**

1. Is the manuscript technically sound, and do the data support the conclusions?

Reviewer #1: Yes

Reviewer #2: Yes

Reviewer #3: Partly

2. Has the statistical analysis been performed appropriately and rigorously? 

Reviewer #1: I Don't Know

Reviewer #2: Yes

Reviewer #3: I Don't Know

3. Have the authors made all data underlying the findings in their manuscript fully available?

Reviewer #1: No

Reviewer #2: Yes

Reviewer #3: Yes

4. Is the manuscript presented in an intelligible fashion and written in standard English?

Reviewer #1: Yes

Reviewer #2: Yes

Reviewer #3: Yes

5. Review Comments to the Author

Reviewer #1: General:

Overall, the manuscript presents evidence that can assist to improve the PMTCT program in the study HFs in Cameroon.

Major comments:

1) About the study design(Line 94-95), ….. a hospital-based cohort-study..

• It is known that cohort study is a type of longitudinal study, which is an approach that follows research participants over a period of time (often many years).

• But in this study, mother-child couples (the cohort) were scheduled for two monitoring assessments (the first assessment was after delivery and the second one at six weeks after delivery). And then, the primary outcomes (VL monitoring, mode of delivery according to VL and early infant diagnosis result) were measured only once, at six weeks. So, as a cohort study, there are no multiple measurers (every two week, three month, ..) of the primary outcomes, especially VL coverage, has been done at defined time intervals

• Hence, you need to define how the follow up was done between 1998-2000 and what parameters (the primary outcomes, or other parameter) has been measured at regular intervals. And you need to show the results in Tabe form. Otherwise, this study might not qualify the study design as “Cohort study”

2) In the result section (Line 199 – 216), There is/are no results (Tables) which support the explanation

Minor comments: showed in the Table below (in attachment)

Reviewer #2: Kafak and colleagues aimed at ascertaining the performance in plasma viral load monitoring during antenatal care, to describe the mode of delivery of HIV infected pregnant women, and to estimate the rate of HIV-1 vertical transmission according to maternal viremia in three facilities of the Littoral region of Cameroon from January 2019 to May 2021. The analysis was performed on 135 women and their babies. Authors found that in the Littoral region, plasma viral load coverage was poor, indicating a suboptimal viral load monitoring among pregnant women living with HIV. Despite this, authors observed an improvement in scaling-up viral load overtime. The manuscript is in general well written and understandable. However, some few aspects should be clarified.

My specific comments are reported below:

1. Results, line 169: Authors should the word “women” after “pregnant”.

2. Results, lines 169-172: Authors should describe the variable “age” in a similar way in both abstract and main text. In fact, this variable is reported as mean in the main text, while in the abstract is reported as median.

3. Results, lines 201 & 202: Findings reported in lines 201&202 are repeated in a more correct way in the lines 209-216. Therefore, in my opinion the following paragraph needs to be deleted: “… with a 201 lower VL-coverage before the year-2020 (p=0.045). Married women (p=0.0273) and those on treatment before pregnancy (p�0.0001) had a higher VL-coverage.

4. Results, line 212: Authors should change”61,8%” into ”61,.8%”.

5. Table 2: Authors should report in a legend the meaning of the acronyms DGH, LDH and NRH. Moreover, in parenthesis, after DGH, LDH, NRH and Total, for each column, they should report the number of women analysed: e.g. Total (N=135).

6. Tables 2 & 3: Some numbers and percentages are in bold. Atuhors should explain why in the legend.

7. Table 3: After DGH, LDH, NRH and Total, for each column, authors should report in parenthesis the number of women analysed: e.g. Total (N=135).

8. Table 3: The percentages of Gestational age at labour in the total population should be reported with only one decimal place.

9. Table 4: What do indicate the asterisks?

Reviewer #3: Reviewer comments

With great interest we read the manuscript entitled : evaluation of plasma viral_load monitoring and the prevention of mother_to_child transmission of HIV_1 in the Littoral Region of Cameroon ».

The mauscript is well presented with results that can impact public health. Nevertheless, authors should address some issues identified.

We will like to suggest that in the title, to replace monitoring by coverage. Also add ..in three health facilities in the Littoral Region. Data obtained may not be applicable in the whole region, given that only 153 participants were enrolled in the study.

Page 1 : line 21 : we did not see two authors with the symbol &

Abstract : In the backgroung….line 26-27 : Replace monitoring by coverage. Materno-fœtal outcomes of pregnancy : this expression is too broad and non specific ; please relate to your findings . What is the outcome of a pregancy ?

Methods : Reading the work, we can not justify it was a cohort study !

Line 30-31 : In more than one year only 135 participants were enrolled in three hospitals ? Can you discuss this poor participation ?

Line 33 : replace outcome by results, and say : EID results were analysed not stratified

Overall as state before, 135 participants can not represent a whole littoral region ! soften your conclusion !

Line 85 : rephrase, state of the art is not appropriate here

Methods

Line 100 : Please rephrase. I think nothing was done for the mother and only the child was tested for HIV infectio at 6 weeks.

Line 112 : give details of PMTCT guidelines in Cameroon. Eg, number of ANC visits recommended…

Line 122 : replace enrolled by selected

Line 127 : Table 1 : clarify the title and instead of writing hospital 1, 2 or 3 spelled out the name of these hospitals

Line 131 : clarify « standard questionnaire »

The methodology is not clairly presented. Read line 132-134 ! two monitoring assessments, what did you assess, what did you monitor What are the study variables ?

Data collection

Line 137-138 rephrase

Data analysis

Line 144 : VL tests occuring ?

Line 152laboratory parameters are which ? specify !

Line 161 : change the term automony, it is not appropriate here

Line 164 : what is non-maleficence ?

Results

Table 2 : define abbreviation at the bottom of the table DGH, LDH, NRH. What is grand multiparous ?

Line 187-190 : not clair ! Please rephrase

Table 3 : the expression fœtal outcomes is not appropriate suggestion : children description, correct the title as well

Line 197-198 : this title is not understood… associated factors to what ?

Line 199 : what is PMTCT patients ? please, revise !

Discussion

Line 237-238 : please revise; have been tested for VL instead of received a VL testing

Line 253 : disclosure of HIV status, and please rephrase the whole sentence for clarity !

6. PLOS authors have the option to publish the peer review history of their article (what does this mean?). If published, this will include your full peer review and any attached files.

Reviewer #1: **Yes: **Desta Kassa

Reviewer #2: No

Reviewer #3: **Yes: **Celine NKENFOU

---

## [Author Response · Author response to Decision Letter 0]

23 Sep 2022

REPONSE TO THE EDITOR’S AND TO REVIEWERS’ COMMENTS

Journal Requirements:

Response 1: The paper has been formatted according to PLOS ONE’s style.

Response 2: informed consent was written. This has been revised in the current version of the manuscript. The revised statement now reads as follows:

“For each study participant, written informed consent was provided; confidentiality in handling the participant information was ensured by the use of specific identifiers; the storage of paper sheets in a looked office, and data entry in an encrypted database on a password-protected computer, and clinical advice or medical care was provided individually to each participant for their personal health benefits.”

3. In your Methods section, please provide additional information on how the sample size was determined and why sample size calculation was not carried out before sample collection.

Response 3: A sub-section has been provided for sample size, as shown below:

Sampling technique and sample size calculation

A consecutive and non-probabilistic sampling method was used to enrol study participants who meet the inclusion criteria (pregnant women who is HIV-seropositive, receiving antiretroviral therapy as per our national guidelines, registered for ANC in one of the study sites, and has provided a written informed consent).

The minimum sample size was calculated using the Cochran’s Formula as follows:

 With n being the minimum sample size; p being the proportion of viral load coverage among HIV-infected pregnant women in Cameroon during the year 2021 (90.7%, i.e. 0.907) [13]; with z being the standard normal random variable at 95% confidence interval (1.96); and d being the error margin (5%, i.e. 0.05). With n = 129.6, the minimum sample size for the study was 130 pregnant women to be enrolled. 

 NA. 

Response 4: We thank the editor for this revision. The following statement has been added:

The study was supported by the government of Cameroon through provision of viral load testing kit and antiretroviral drugs to all the study participants. “The funders had no role in study design, data collection and analysis, decision to publish, or preparation of the manuscript.”.

Additional Editor Comments:

There are some issues with the design of the study/cohort, as shown as a major concern by one of the reviewers, should be redesigned and addressed. The inclusion of the subjects should be in a systematic manner to provide meaningful conclusions.

Response to additional editor comments: the study design has been revised accordingly. Thus, “cohort” has been removed from the statement, and the current design is a hospital-based study.

Review Comments to the Author

Reviewer #1: General:

Overall, the manuscript presents evidence that can assist to improve the PMTCT program in the study HFs in Cameroon.

Major comments:

1) About the study design(Line 94-95), ….. a hospital-based cohort-study..

• It is known that cohort study is a type of longitudinal study, which is an approach that follows research participants over a period of time (often many years).

• But in this study, mother-child couples (the cohort) were scheduled for two monitoring assessments (the first assessment was after delivery and the second one at six weeks after delivery). And then, the primary outcomes (VL monitoring, mode of delivery according to VL and early infant diagnosis result) were measured only once, at six weeks. So, as a cohort study, there are no multiple measurers (every two week, three month, ..) of the primary outcomes, especially VL coverage, has been done at defined time intervals

• Hence, you need to define how the follow up was done between 1998-2000 and what parameters (the primary outcomes, or other parameter) has been measured at regular intervals. And you need to show the results in Table form. Otherwise, this study might not qualify the study design as “Cohort study”.

Response to comment 1: We agree with the reviewer that all the study primary outcomes were measured just once (except for viral load coverage which was evaluated during each trimester of the ANC, as shown in the revised table 4). Taking this into consideration, the statement “cohort” has been deleted and the study design has been revised to a hospital-based study.

2) In the result section (Line 199 – 216), There is/are no results (Tables) which support the explanation

Response to comment 2: We thank the reviewer for this observation. We have provided a detailed table supporting the information shown in the text within this section.

Minor comments: showed in the Table below (in attachment)

Response to minor comment 1: We have considered all the suggestions raised by the reviewer.

Reviewer #2: Kafak and colleagues aimed at ascertaining the performance in plasma viral load monitoring during antenatal care, to describe the mode of delivery of HIV infected pregnant women, and to estimate the rate of HIV-1 vertical transmission according to maternal viremia in three facilities of the Littoral region of Cameroon from January 2019 to May 2021. The analysis was performed on 135 women and their babies. Authors found that in the Littoral region, plasma viral load coverage was poor, indicating a suboptimal viral load monitoring among pregnant women living with HIV. Despite this, authors observed an improvement in scaling-up viral load overtime. The manuscript is in general well written and understandable. However, some few aspects should be clarified.

My specific comments are reported below:

1. Results, line 169: Authors should the word “women” after “pregnant”.

Response 1: the word has been added as indicated by the reviewer.

2. Results, lines 169-172: Authors should describe the variable “age” in a similar way in both abstract and main text. In fact, this variable is reported as mean in the main text, while in the abstract is reported as median.

Response 2: the age has been revised to median in the result section as indicated by the reviewer.

3. Results, lines 201 & 202: Findings reported in lines 201&202 are repeated in a more correct way in the lines 209-216. Therefore, in my opinion the following paragraph needs to be deleted: “… with a 201 lower VL-coverage before the year-2020 (p=0.045). Married women (p=0.0273) and those on treatment before pregnancy (p�0.0001) had a higher VL-coverage.

Response 3: the section has been revised and the specific statement deleted as indicated by the reviewer.

4. Results, line 212: Authors should change”61,8%” into ”61,.8%”.

Response 4: this has been revised as indicted by the reviewer.

5. Table 2: Authors should report in a legend the meaning of the acronyms DGH, LDH and NRH. Moreover, in parenthesis, after DGH, LDH, NRH and Total, for each column, they should report the number of women analysed: e.g. Total (N=135).

Response 5: the meaning of each acronym and the number of women analysed have been provided as required by the reviewer.

6. Tables 2 & 3: Some numbers and percentages are in bold. Authors should explain why in the legend.

Response 6: numbers and percentages in bold have been explained accordingly, as required by the reviewer.

.

7. Table 3: After DGH, LDH, NRH and Total, for each column, authors should report in parenthesis the number of women analysed: e.g. Total (N=135).

Response 7: the number of women analysed has been added consistently.

8. Table 3: The percentages of Gestational age at labour in the total population should be reported with only one decimal place.

Response 8: the decimal has been revised accordingly.

9. Table 4: What do indicate the asterisks?

Response 9: the asterisks indicate statistically significant values. This has been added accordingly in the legend.

Reviewer #3: Reviewer comments

With great interest we read the manuscript entitled : evaluation of plasma viral_load monitoring and the prevention of mother_to_child transmission of HIV_1 in the Littoral Region of Cameroon ».

The mauscript is well presented with results that can impact public health. Nevertheless, authors should address some issues identified.

1. We will like to suggest that in the title, to replace monitoring by coverage. Also add ..in three health facilities in the Littoral Region. Data obtained may not be applicable in the whole region, given that only 135 participants were enrolled in the study.

Response 1: we fully agree with the reviewer’s suggestions. These have been effected in the revised version of the manuscript. 

2. Page 1 : line 21 : we did not see two authors with the symbol &

Response 2: this symbol attribute has been revised accordingly.

3. Abstract : In the background….line 26-27 : Replace monitoring by coverage. Materno-fœtal outcomes of pregnancy : this expression is too broad and non specific ; please relate to your findings . What is the outcome of a pregancy ?

Response 3: monitoring have been replaced by coverage. For better clarity, and due to limited wording in the abstract, materno-foetal outcomes has been explained in the full text, as required by the reviewer (see sentence added under data analysis section of the methodlogy). 

4. Methods : Reading the work, we can not justify it was a cohort study !

Response 4: we have deleted the word “cohort” from the study design, as required. 

5. Line 30-31 : In more than one year only 135 participants were enrolled in three hospitals ? Can you discuss this poor participation ?

Response 5 : we thank the reviewer for this comment, the following statement has been added in the discussion section of the paper. 

“In spite of the high number of women of child bearing age and pregnant women in our setting, the sample size (n=135 HIV-pregnant women enrolled) during the study period correlates to the reducing burden of HIV-infection in ANC [13].”

6. Line 33 : replace outcome by results, and say : EID results were analysed not stratified

Response 6 : these revisions have been done.

7, Overall as state before, 135 participants can not represent a whole littoral region ! soften your conclusion !

Response 7 : the conclusion has been soften as recommended by the reviewer. 

8. Line 85 : rephrase, state of the art is not appropriate here

Response 8 : the statement has been revised to “effective implementation". 

Methods

9. Line 100 : Please rephrase. I think nothing was done for the mother and only the child was tested for HIV infectio at 6 weeks.

Response 9: The statement has been revised as follows: “infants were followed-up to six weeks postnatal for HIV early infant diagnosis (EID)”.

10. Line 112 : give details of PMTCT guidelines in Cameroon. Eg, number of ANC visits recommended…

Response 10: These statements have been added as required acordingly.

11. Line 122 : replace enrolled by selected.

Response 11: This has been done accordingly.

12. Line 127 : Table 1 : clarify the title and instead of writing hospital 1, 2 or 3 spelled out the name of these hospitals.

Response 12: This has been done accordingly.

13. Line 131 : clarify « standard questionnaire ». The methodology is not clairly presented. Read line 132-134 ! two monitoring assessments, what did you assess, what did you monitor What are the study variables 

Response 13: Clarity has been provided in the text and we have changed “standard questionnaire” to read “structured questionnaire”.

14. Data collection, Line 137-138 rephrase

Response 14: the rephrasing has been done accordingly.

15. Data analysis, Line 144 : VL tests occuring ?

Response 15: this has been rephrased to “All VL results during pregnancy”

16. Line 152 laboratory parameters are which ? Specify 

Response 16: this has been specified.

17. Line 161 : change the term automony, it is not appropriate here

Response 17: this has been rephrased.

L18. Line 164 : what is non-maleficence ?

Response 18: this has been deleted, since not essential to mention. 

Results

19. Table 2 : define abbreviation at the bottom of the table DGH, LDH, NRH. What is grand multiparous ?

Response 19: the input has been provided accordingly.

20. Line 187-190: not clair ! Please rephrase.

Response 20: this section has been rephrased to read:

“Looking at the obstetrics information of HIV-infected pregnant women, 92.6% (125/135) had spontaneous labour and the mean gestational age at labour was 38,55±1.74SD; rupture of membranes (ROM) was 91.1%; and the population of HIV-exposed children were composed of more male than female (55.6% vs 44.4% respectively) and 89.6% had a normal weight (Table 3).”

21. Table 3 : the expression fœtal outcomes is not appropriate suggestion : children description, correct the title as well.

Response 21: this section has been rephrased accordingly

22. Line 197-198 : this title is not understood… associated factors to what ?

Response 21: this title has been rephrased accordingly

23. Line 199 : what is PMTCT patients ? please, revise !

Response 23: this sentence has been rephrased accordingly.

24. Discussion. Line 237-238 : please revise; have been tested for VL instead of received a VL testing

Line 253 : disclosure of HIV status, and please rephrase the whole sentence for clarity.

Response 24: these sentences have been rephrased accordingly.

Section by section questions/Comments/suggestions 

Section Line No and Concerns highlighted in red font Questions/Comments/suggestions

Title 

Abstract-Result Line 36-37: VL-coverage during antenatal care (ANC) was 50.4% (68/135), with a lower VL coverage before the year-2020 (37.5% v. 61.9%, p=0.0069). ….before the year 2020 … is an open end. Could it be with the study period (between 2019 to 2022)?

Response: the study period has been revised accordingly.

 Line 37-38: Married women (p=0.0273) and those on treatment before pregnancy (p<0.0001) had a higher VL coverage. • Indicate the comparison group in each: Married women vs…; those on treatment vs….

• “Higher” is subjective --- better to be supported with figures (%)

Response: this has been considered accordingly.

 Line 40: … suggesting an overall viral suppression of 89.7% (61/68). • … over all viral suppression…… can you include (<= 1000 copies/ml)

Response: this has been considered accordingly.

Conclusion Line 45-46: In these Cameroonian settings, VL-coverage remains suboptimal among ANC attendees, • Your reference? National coverage, national target, or global target? Please indicate (? %)

Response: this has been provided.

 Line 46-47: Viral suppression rate remains below the target for eliminating MTCT • …. Below the target for eliminating MTCT ….. national or global target, please specify, and indicate the target in bracket (? %) 

• Response: this has been provided accordingly.

 Line 48-49: Thus, achieving an optimal PMTCT performance requires a thorough compliance to virologic assessment during ANC. • This conclusion needs to be supported by findings showed in the result section. i.e. Was routing VL testing for the HIV Pos pregnant women has not been done as per the national/global guideline during the study period-2019-2022 ?

Response: this has been considered accordingly, and the conclusion has been soften.

Study population Enrolment of study participants Any exclusion criteria? Please indicate

Response: this has been provided in the methodology section.

Results Line 200-201…with a lower VL-coverage before the year-2020 Please qualify “Lower” number ? (%); and also qualify “before the year 2020”, does it mean between 1998-2000 ?

Response: this has been clarified in the text.

 Line 201-202: …..Married women (p=0.0273) and those on treatment before pregnancy (p�0.0001) had a higher VL-coverage Please show, with whom the “Married women” and those on treatment” are compared and indicate the figures (%) like ---% vs ….. (p=0.0273, for both comparisons

Response: this has been considered through out.

 Line 199-2016 …. There is/are no results (Tables) which support the explanation 

Response: this has been revised accordingly.

Discussion Overall, the results are well discussed!!

 Line 242 … regarding your discussion on VL coverage, you need to show your results in Table in the result section, then you can discuss your findings. Overall, you better show in Table (s) of the results regarding VL coverage over time (during the study period)

Response: the table has been provided as indicated.

 Line 246-247 ….Unlike recommendations from our national guidelines on PMTCT Pleas show what the national guideline recommends.

Response: this has been provided accordingly.

Conclusion Line 324 ..Nonetheless, there is an improvement in scaling-up VL overtime, As commented earlier, VL coverage overtime need to be showed in the result section in a Table.

Response: this has been provided as aforementioned.

---

## [Decision Letter · Decision Letter 1]

25 Oct 2022

Evaluation of Plasma Viral-Load Monitoring and the Prevention of Mother-To-Child Transmission of HIV-1 in three health facilities of the Littoral Region of Cameroon

PONE-D-22-15160R1

Dear Dr. Fokam,

We’re pleased to inform you that your manuscript has been judged scientifically suitable for publication and will be formally accepted for publication once it meets all outstanding technical requirements.

Kind regards,

Nafees Ahmad, Ph.D.

Academic Editor

PLOS ONE

Additional Editor Comments (optional):

Reviewers' comments:

Reviewer's Responses to Questions

**Comments to the Author**

1. If the authors have adequately addressed your comments raised in a previous round of review and you feel that this manuscript is now acceptable for publication, you may indicate that here to bypass the “Comments to the Author” section, enter your conflict of interest statement in the “Confidential to Editor” section, and submit your "Accept" recommendation.

Reviewer #1: All comments have been addressed

Reviewer #2: All comments have been addressed

Reviewer #3: All comments have been addressed

2. Is the manuscript technically sound, and do the data support the conclusions?

Reviewer #1: Yes

Reviewer #2: Yes

Reviewer #3: Yes

3. Has the statistical analysis been performed appropriately and rigorously? 

Reviewer #1: I Don't Know

Reviewer #2: Yes

Reviewer #3: Yes

4. Have the authors made all data underlying the findings in their manuscript fully available?

Reviewer #1: Yes

Reviewer #2: Yes

Reviewer #3: Yes

5. Is the manuscript presented in an intelligible fashion and written in standard English?

Reviewer #1: Yes

Reviewer #2: Yes

Reviewer #3: (No Response)

6. Review Comments to the Author

Reviewer #1: (No Response)

Reviewer #2: By answering to all the reviewers' comments, authors definitely improved the quality of the manuscript. Therefore, now in my opinion the manuscript is suitable as a publication in the PLOSONE journal.

Reviewer #3: With satisfaction, I read the revised manuscript. All my comments have been addressed. I have identified a number of typos. I invite the authors to correct them!

7. PLOS authors have the option to publish the peer review history of their article (what does this mean?). If published, this will include your full peer review and any attached files.

Reviewer #1: No

Reviewer #2: No

Reviewer #3: **Yes: **Celine N Nkenfou
